# Random Propagations in GNNs

**Thu Bui[1], Anugunj Naman[1], Carola-Bibiane Schönlieb[2],**
**Bruno Ribeiro[1], Beatrice Bevilacqua[1], Moshe Eliasof[2]**
[1]Department of Computer Science, Purdue University, West Lafayette, IN, USA
[2]Department of Applied Mathematics, University of Cambridge, Cambridge, UK
{bui35, anaman, ribeirob, bbevilac}@purdue.edu
{cbs31, me532}@cam.ac.uk

## Abstract

Graph learning benefits many fields. However, Graph Neural Networks (GNNs) often struggle with scalability, especially on large graphs. At the same time, many tasks seem to be simple in terms of learning, e.g., simple diffusion yields favorable performance. In this paper, we present **Ra**ndom **P**ropagation **GNN** (RAP-GNN), a framework that addresses two main research questions: (i) can random propagations in GNNs be as effective as end-to-end optimized GNNs? and (ii) can they reduce the computational burden required by traditional GNNs? Our empirical findings indicate that RAP-GNN reduces training time by up to 58%, while maintaining strong accuracy for node and graph classification tasks.

## 1 Introduction

Graph learning has become crucial across fields like biology and recommendation systems, with Graph Neural Networks (GNNs) at the core [1]. While GNNs perform well on various tasks, their computational demands can be high due to backpropagation through every layer, making large-scale training costly. Early studies introduced lightweight alternatives, such as diffusion [2, 3], linear GNNs [4], and neighborhood sampling [5]. More recently, Forward-Forward learning [6] bypassed backpropagation by training layers independently, including in GNNs [7, 8]. However, this still requires training each layer and computing outputs for all classes. Random propagation in GNNs, which uses random weights instead of backpropagation, offers a more efficient approach, significantly reducing training time and computational costs. This method allows the processing of deeper, larger graphs without significant performance loss [9–11], though it has been mostly limited to simpler tasks and GNN architectures.

In this work, we address the following questions:

1. How effective is random propagation across various datasets and tasks, from small to large graphs?

2. Can random weights, requiring only the final classifier to be trained, serve as a viable alternative to current methods?

To explore this, we introduce **Ra**ndom **P**ropagation **GNN** (RAP-GNN), a framework utilizing randomly sampled weights in GNN layers and a pretrained feature embedding, requiring only the classifier to be trained. Our experiments on node and graph level tasks show that RAP-GNN cuts training time by up to 58% while maintaining competitive accuracy compared to end-to-end trained GNNs.

Preprint.

## 2 Related Work

Several methods have been proposed to improve GNN training efficiency, as outlined below.

**Backpropagation-Free Training Methods.** The Forward-Forward approach simplifies training by avoiding backpropagation and training each layer independently. Initially applied to computer vision [6], it was later adapted to GNNs [7, 8]. However, training all layers and computing outputs for all classes is still required. In contrast, our method only trains the classifier, using random propagation through the GNN layers, which reduces computational overhead.

**Graph Lottery-Ticket Hypothesis.** The Graph Lottery-Ticket approach proposed recently [12, 13] suggests that sparse sub-networks can perform as well as fully trained models. However, this method requires pretraining the entire network to find these sub-networks, which remains computationally intensive.

**Random Models.** The most closely related work comes from Reservoir Computing (RC), where fixed, randomly sampled reservoirs capture graph dynamics without extensive training [14, 9, 15]. Traditional RC methods [14, 9] rely on recurrent forward passes until convergence or a set number of iterations, which can be computationally demanding. FDGNN [9] introduced a GNN framework with random, fixed weights for graph classification, using stability constraints in a recurrent setting. MRGNN [15] extended this by "unrolling" the recurrent hidden layer, reducing time complexity. These methods rely on static random weights and are primarily suited for graph classification, yet demonstrate strong performance at lower computational cost. GCN-RW [10] further improved efficiency with random filters optimized through least squares, enabling faster training in node classification tasks without sacrificing accuracy. Moreover, randomness in node features [16–19] and its propagation has proven effective as a positional encoding technique [20] and within the normalization layer [21]. Similarly, Yu et al. [22] explored how adding noise to graph features can improve performance, paralleling the use of random initialization as a form of augmentation. Although the method does not focus on improving training efficiency, it underscores the versatility of randomness as a tool for improving efficiency and performance in GNN tasks.

Additionally, recent work [23, 24] questions the necessity of graph convolution during training, suggesting that alternative methods, such as post-training modifications (e.g., Graph-ODE), can achieve strong results. They highlight the potential for bypassing training of GNN layers, aligning with the random weight techniques used in our method. Our RAP-GNN, which dynamically samples random weights in each hidden GNN layer during every forward pass, similarly demonstrates that full reliance on GNN may not be essential for achieving strong performance. This method enhances representation flexibility while reducing computational overhead.

## 3 RAP-GNN

In this section, we introduce RAP-GNN, which uses randomness in GNN layers while only training the final classifier. Random weights for all GNN layers are sampled from a uniform distribution at each forward pass, eliminating backpropagation and significantly reducing computational costs. We also utilize a pretrained feature embedding to process input node features. The necessity of this pretrained embedding is discussed in Appendix C. The full architecture of RAP-GNN variants using randomly sampled GNN weights is shown in Figure 1, with other variants detailed in Section 4.

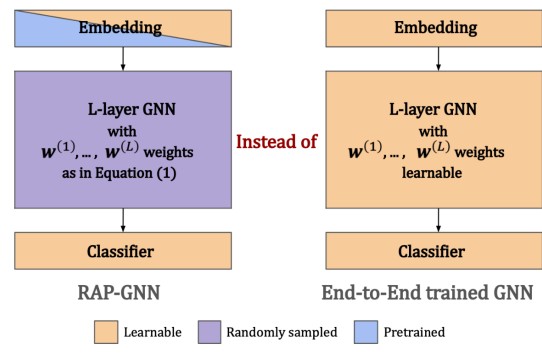

Figure 1: RAP-GNN Framework: We employ an L-layer GNN with randomly sampled GNN weights as defined in Equation (1) instead of learnable GNN weights, paired with either a pretrained or learnable embedding.

The model RAP-GNN consists of three components: (i) an embedding layer $h_\phi^{\text{pre}}$, (ii) a stack of GNN layers with non-linearities $g_{\mathbf{W}}$, and (iii) a classifier $c_\theta$. The embedding layer $h_\phi^{\text{pre}} : \mathbb{R}^p \rightarrow \mathbb{R}^d$ maps input feature dimension, p, into a hidden-dimensional space, d, using a multi-layer perceptron (MLP) or a single-layer

GNN. The GNN layers $g_{\mathbf{W}} : \mathbb{R}^{n \times d} \times \mathbb{R}^{n \times n} \rightarrow \mathbb{R}^{n \times d}$ process these representations, with weights $\mathbf{W} = [\mathbf{w}^{(1)}, \mathbf{w}^{(2)}, \ldots, \mathbf{w}^{(L)}]$ sampled uniformly at each forward pass. If $h_\phi^{\text{pre}}$ is learnable, backpropagation through the GNN layers would be needed, but this is avoided with a pretrained embedding. The classifier $c_\theta : \mathbb{R}^d \rightarrow \mathcal{Y}$ maps representations to the target space $\mathcal{Y}$, with only the classifier parameters $\theta$ learned during training. The model is trained in two phases:

**Pretraining Phase.** The embedding layer, $h_\phi^{\text{pre}}$, is pretrained on the dataset for the downstream task using a simplified network, $f^{\text{pre}} = c_{\phi'}^{\text{pre}} \circ h_\phi^{\text{pre}}$. This step optimizes the embedding layer to extract meaningful features from the input data. By pretraining the embedding layer, the model captures relevant features early on, eliminating the need to update $\phi$ during the main training phase.

Appendix C empirically demonstrates that using dynamic random embeddings at each call introduces excessive noise, while fixed random embeddings lack adaptability during training, both of which degrade accuracy. Therefore, this pretraining step is essential in scenarios where extensive training is undesirable or impractical.

**Training Phase.** In the training phase, the key innovation is introducing randomness into the GNN network, $g_{\mathbf{W}}$, which consists of $L$ hidden layers. During each forward pass, a new random diagonal weight matrix $\mathbf{w}^{(l)}$ is sampled for each GNN layer. In Appendix A, we detail how $\mathbf{w}^{(l)}$ is applied across different GNN backbones, but it is generated as follows:

$$
\mathbf{w}^{(l)} = \begin{pmatrix} \alpha_1^{(l)} & & \mathbf{0} \\ & \ddots & \\ \mathbf{0} & & \alpha_{\text{d}}^{(l)} \end{pmatrix} \tag{1}
$$

where $\boldsymbol{\alpha}^{(l)} = [\alpha_1^{(l)}, \ldots, \alpha_d^{(l)}]$ is a vector randomly sampled from the uniform distribution, with d = hidden dims. The diagonal entries $\alpha_i^{(l)}$ are constrained to the interval $[0, 1]$. Thus, $\boldsymbol{\alpha}^{(l)}$ is sampled from a uniform distribution $U(0, 1)$. The diagonal structure ensures that each feature is propagated independently by the corresponding $\alpha_i^{(l)}$, thereby preventing full random weights from mixing up the features. Limiting each $\mathbf{w}^{(l)}$ to include values in [0,1], combined with GCN [25], renders diffusion propagations, as shown in Eliasof et al. [26]. The main difference is that instead of learning $\mathbf{w}^{(l)}$, it is randomly sampled at each forward pass of the network. This controlled propagation type allows the model to explore a rich variety of feature representations while avoiding excessive disruption of the learned structure.

During backpropagation in this training phase, only the parameters $\theta$ of the classifier $c_\theta$ are updated based on the loss function, while $\mathbf{W}$ and $\phi$ remain unchanged. This eliminates the need for backpropagation through the GNN layers, significantly reducing computational costs.

**Inference with Majority Voting.** During inference, majority voting is employed to enhance robustness and generalization, with the number of votes, M, treated as a hyperparameter. For each vote $j \in \{1, \ldots, M\}$, a new random vector $\boldsymbol{\alpha}^{(l)}$ is sampled to generate a corresponding $\mathbf{w}^{(l)}, \forall l \in [L]$, as outlined in Equation (1). The model then computes the output $\hat{y}_j = f(\mathbf{x}^{\text{te}})$ for each vote , where $\mathbf{x}^{\text{te}}$ is the testing data. The final prediction $\hat{y}$ is determined by majority voting across all voters' outputs $\hat{y}_j$. A similar scheme was used by Bevilacqua et al. [27] to reduce stochasticity in subgraph sub-sampling.

Our approach leverages on-the-fly random sampling of diagonal weights for all GNN layers, combined with a fixed pretrained embedding layer and a trainable classifier. This enhances generalization across diverse graph structures while improving robustness and efficiency, as empirically validated in Section 4 and Appendix D. The pseudocode for the algorithm is provided in Appendix A.

## 4   Experiments

We empirically address our key questions by first evaluating whether training the entire network is necessary or if randomness in GNN layers can serve as an effective alternative. We assess the impact of different randomness strategies on downstream performance using the Cora [28], CiteSeer [29],

Table 1: RAP-GNN accuracy performance (%)↑ with a GCN backbone on node classification is evaluated against two end-to-end networks. The results show that RAP-GNN achieves *competitive accuracy*, even can *surpass* the baselines across various datasets with LEARNEMB.

| Method ↓ / Dataset → | CORA | CITESEER | PUBMED | OGBN-ARXIV |
|---|---|---|---|---|
| **NATURAL BASELINES** | | | | |
| LW | $81.50 \pm 0.8$ | $71.10 \pm 0.7$ | $79.00 \pm 0.6$ | $\mathbf{73.41 \pm 0.2}$ |
| LW + LEARNEMB + RNF | $81.25 \pm 0.6$ | $70.34 \pm 0.2$ | $78.83 \pm 0.7$ | $73.28 \pm 0.3$ |
| **RAP-GNN VARIANTS** | | | | |
| IPW + LEARNEMB | $82.28 \pm 0.6$ | $\mathbf{73.58 \pm 0.2}$ | $78.86 \pm 0.3$ | $70.57 \pm 0.1$ |
| IPW + LEARNEMB + RNF | $81.86 \pm 0.5$ | $73.64 \pm 0.3$ | $79.14 \pm 0.2$ | $70.59 \pm 0.2$ |
| FPW + LEARNEMB | $81.92 \pm 0.9$ | $70.98 \pm 0.5$ | $78.34 \pm 0.5$ | $70.28 \pm 0.6$ |
| FPW + LEARNEMB + RNF | $82.34 \pm 0.8$ | $71.56 \pm 0.3$ | $79.13 \pm 0.2$ | $70.15 \pm 0.4$ |
| DPW + LEARNEMB | $82.82 \pm 0.9$ | $71.48 \pm 0.7$ | $78.76 \pm 0.6$ | $70.58 \pm 0.3$ |
| DPW + LEARNEMB + RNF | $\mathbf{84.36 \pm 0.3}$ | $72.16 \pm 0.6$ | $\mathbf{79.32 \pm 0.3}$ | $71.23 \pm 0.2$ |
| DPW + PREEMB + RNF | $83.33 \pm 0.2$ | $70.80 \pm 0.3$ | $78.70 \pm 0.4$ | $69.84 \pm 0.4$ |

PubMed [30] and ogbn-arxiv [31] datasets. Additionally, we compare the training and evaluation times of these strategies with those of end-to-end backpropagation, using a GCN backbone [25].

For further evaluation across tasks and backbones, Appendix D provides results for graph classification on TUDatasets [32] with a GIN backbone [33], and additional results on ogbn-arxiv using GraphSage [5]. We also assess the performance of RAP-GNN using different randomness strategies in the embedding layer versus a pretrained embedding on Cora, CiteSeer, and PubMed. Full implementation details are in the Appendix E.

## 4.1 Impact of Randomness in GNN layers

We evaluate the impact of three randomness strategies in GNN weights: (i) Identity Propagation Weights (IPW): All $\mathbf{w}^{(l)}$ in $\mathbf{W}$ are identity matrices, passing input representations unchanged, without using randomly sampled weights. (ii) Fixed Propagation Weights (FPW): All $\mathbf{w}^{(l)}$ are randomly initialized and fixed throughout. (iii) Dynamic Propagation Weights (DPW): All $\mathbf{w}^{(l)}$ are randomly sampled at each forward pass. Each strategy is tested with and without random features (RNF). To optimize accuracy, we combine these strategies with a learnable embedding (LEARNEMB) trained via backpropagation, comparing against a standard GCN with fully learnable weights (LW). We further evaluate a RAP-GNN variant that combines a pretrained embedding (PREEMB) with DPW and RNF. Additionally, we introduce another trained end-to-end baseline combining LW, LEARNEMB and RNF. We use majority voting with voter count M = 5 for the Cora, Citeseer, and PubMed datasets, and M = 3 for ogbn-arxiv in the RAP-GNN variants.

Firstly, without prioritizing runtime optimization, we focus on evaluating how different randomized sampling methods combined with LEARNEMB performs compared to fully end-to-end trained networks. The results in Table 1 reveal consistent trends across the Cora, CiteSeer, and PubMed datasets. Among the end-to-end baselines, the LEARNEMB setup slightly underperforms compared to baseline without LEARNEMB, likely due to the less expressive MLP backbone used in LEARNEMB compared to GCN. However, within the variations of RAP-GNN combined with LEARNEMB, IPW achieves accuracy comparable to fully end-to-end models on all three datasets. Even FPW, which limits random sampling in GNN, reaches the same accuracy levels as the end-to-end baselines. Notably, DPW combined with LEARNEMB consistently improves over the end-to-end networks across these datasets, highlighting the strength of leveraging randomness in GNN layers, which can even surpass traditional fully learned networks.

On the more challenging ogbn-arxiv dataset, all variations of RAP-GNN combined with LEARNEMB exhibit only a slight decrease in accuracy compared to the two end-to-end baselines, further demonstrating that randomized GNN layers can still deliver competitive results.

We also notice that DPW with a pretrained embedding show a slight accuracy drop compared to a learnable embedding across all datasets (since the embeddings aren't updated during training), they still outperform the end-to-end trained baselines on Cora. Although these are early-stage results,

Table 2: Running time for training (TRAIN) and evaluation (TEST) with varying voter counts (M) using a GCN backbone for node classification. RAP-GNN with PREEMB *reduces training time* by *58%*.

| Method ↓ / Dataset → | CORA (ms) | | | OGB-ARXIV (ms) | | |
|---|---|---|---|---|---|---|
| | TRAIN | TEST (M=5) | TEST (M=1) | TRAIN | TEST (M=3) | TEST (M=1) |
| **NATURAL BASELINES** | | | | | | |
| LW + LEARNEMB | 4.89 | – | **0.31** | 438.78 | – | 214.28 |
| LW + LEARNEMB + RNF | 5.10 | – | 0.39 | 441.63 | – | 214.70 |
| **RAP-GNN VARIANTS** | | | | | | |
| IPW + LEARNEMB | 2.36 | 2.76 | 0.34 | 397.43 | 644.02 | 214.73 |
| IPW + LEARNEMB + RNF | 2.44 | 3.61 | 0.36 | 425.27 | 689.64 | 229.84 |
| FPW + LEARNEMB | 2.42 | **2.38** | 0.48 | 373.29 | **643.60** | **214.54** |
| FPW + LEARNEMB + RNF | 2.21 | 2.79 | 0.55 | 339.15 | 688.94 | 229.61 |
| DPW + LEARNEMB | 2.88 | 3.68 | 0.74 | 408.16 | 659.64 | 219.88 |
| DPW + LEARNEMB + RNF | 2.56 | 3.96 | 0.80 | 439.86 | 711.59 | 237.17 |
| DPW + PREEMB + RNF | **2.04** | 3.98 | 0.89 | **228.59** | 660.89 | 220.31 |

they underscore the potential of our approach for real-world applications where both accuracy and efficiency are critical.

## 4.2 Time Analysis

To assess the efficiency of RAP-GNN, we report the average runtime for a single training epoch (TRAIN) and on the whole test (TEST) on the small Cora dataset (with M = 5 votes) and the larger ogbn-arxiv dataset (with M = 3 votes). For a fair comparison, all models are configured with the same number of hidden channels, layers, and random features (for RNF models).

In Table 2, we demonstrate that RAP-GNN offers a significant advantage in training efficiency. On Cora, using PREEMB and DPW reduces training time by 58% compared to end-to-end training, as we only need to backpropagate through the classifier. Similarly, on ogbn-arxiv, using PREEMB and DPW reduces training time by 47%. The pretraining times, shown in Appendix D.4, reveal that even when combined with pretraining, the total training time for PREEMB and DPW remains substantially lower than methods with fully learnable weights. Variants using LEARNEMB also show reduced training times, particularly with IPW and FPW, where layer weights are fixed.

However, during evaluation, all RAP-GNN variants experience slower performance due to the need for majority voting (M>1). When M=1 (i.e., without majority voting), the evaluation time is comparable, although DPW lags slightly due to random sampling at each forward pass. Accuracy comparisons for both with and without majority voting on the Cora and ogbn-arxiv datasets are provided in Appendix D.3, suggesting that further investigation is needed into the impact of majority voting.

## 5 Conclusion

We demonstrate that RAP-GNN achieves competitive accuracy in node and graph classification tasks across various small- and large-scale datasets with different GNN backbones while notably reducing training time. These findings highlight that the key component of RAP-GNN, random sampling GNN weights, offers an effective and efficient alternative to end-to-end trained models.

## Acknowledgments and Disclosure of Funding

BR acknowledges support from the National Science Foundation (NSF) awards, CCF-1918483, CAREER IIS-1943364 and CNS-2212160, Amazon Research Award, AnalytiXIN, and the Wabash Heartland Innovation Network (WHIN), Ford, NVidia, CISCO, and Amazon. Computing infrastructure was supported in part by CNS-1925001 (CloudBank). This work was supported in part by AMD under the AMD HPC Fund program. ME is funded by the Blavatnik-Cambridge fellowship, Cambridge Accelerate Programme for Scientific Discovery, and the Maths4DL EPSRC Programme.

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

# A  Method Algorithm

In this section, we outline the steps in RAP-GNN, in Algorithm 1.

---

**Algorithm 1** Pretraining, Training and Evaluation Procedure in RAP-GNN

---

1: **Initialize Model:** $f = c_\theta \circ g_{\mathbf{W}} \circ h_\phi^{\text{pre}}$
- Feature embedding: $h_\phi^{\text{pre}} : \mathbb{R}^{\text{p}} \to \mathbb{R}^{\text{d}}$
    - A MLP or 1-hidden GNN layer
    - p is node feature dimension
    - d is the chosen hidden dimension for GNN layers
- Random-weight GNN Embedding: $g_{\mathbf{W}} : \mathbb{R}^{\text{n} \times \text{d}} \times \mathbb{R}^{\text{n} \times \text{n}} \to \mathbb{R}^{\text{n} \times \text{d}}$
    - GNN with $L$ hidden layers and random weight of each layer denotes as $\mathbf{w}^{(l)}$
- MLP classifier: $c_\theta : \mathbb{R}^{\text{d}} \to \mathcal{Y}$
    - $\mathcal{Y}$ is the target space

2: **Pretraining Phase for $h_\phi^{\text{pre}}$**
3: Initialize learning rate $\eta^{\text{pre}}$ and max epochs $T^{\text{pre}}$ for the pretraining process.
4: **for** $i = 1$ to $T^{\text{pre}}$ **do**
5:     **for** $\mathbf{x}^{\text{tr}}$ in Data$^{\text{tr}}$ **do**         ▷ Data$^{\text{tr}}$ is training dataset.
6:         $\hat{y} = c_{\phi'}^{\text{pre}} \circ h_\phi^{\text{pre}}(\mathbf{x}^{\text{tr}})$
7:         Compute the downstream task loss between the ground-truth $y$ and $\hat{y}$.
8:         Update $c_{\phi'}^{\text{pre}}$ and $h_\phi^{\text{pre}}$ parameters using a gradient-descent method (Adam) with a learning rate $\eta^{\text{pre}}$.
9:     **end for**
10: **end for**

11: **Training Phase**
12: Initialize learning rate $\eta$ and max epochs $T$ for training the model using full standard training data.
13: **for** $i = 1$ to $T$ **do**
14:     **for** $\mathbf{x}^{\text{tr}}$ in Data$^{\text{tr}}$ **do**         ▷ Data$^{\text{tr}}$ is training dataset.
15:         Sample $L$ random vectors $\boldsymbol{\alpha}^{(l)} \in [0,1]^{\text{d}}$.
16:         Generate random GNN weight matrices $\mathbf{w}^{(l)}$ using $\alpha^{(l)}$.     ▷ Equation (1)
17:         Compute model output $\hat{y} = f(\mathbf{x}^{\text{tr}})$.
18:         Update $c_\theta$ parameters using a gradient-descent method (Adam) with learning rate $\eta$.
19:     **end for**
20: **end for**

21: **Evaluation Phase**
22: Set the number of voters $M$
23: **for** $j = 1$ to $M$ **do**
24:     **for** $\mathbf{x}^{\text{te}}$ in Data$^{\text{te}}$ **do**         ▷ Data$^{\text{te}}$ is testing dataset.
25:         Sample $L$ random vectors $\boldsymbol{\alpha}^{(l)} \in [0,1]^{\text{d}}$.
26:         Generate random GNN weight matrices $\mathbf{w}^{(l)}$ using $\alpha^{(l)}$.     ▷ Equation (1)
27:         Compute the model output $\hat{y}_j = f(\mathbf{x}^{\text{te}})$.
28:     **end for**
29: **end for**
30: Compute final prediction $\hat{y}$ via majority voting over $\{\hat{y}_1, \hat{y}_2, \ldots, \hat{y}_M\}$.

---

Each $\mathbf{w}^{(l)}$ functions as a GNN weight matrix, but its application depends on the specific GNN backbone used. Below, we illustrate how $\mathbf{w}^{(l)}$ is utilized in two GNN backbones: GCN and GraphSAGE, both with $L$ hidden layers. For generalization, consider the transition from the $(l-1)$-th layer to the

$l$-th layer, where $\mathbf{x}^{(l-1)}$ represents the output from $(l-1)$-th layer, also the input of the $l$-th layer, while $\mathbf{x}^{(l)}$ is the output of the $l$-th layer).

**GCN.** The forward pass formula for the $l$-th GCN layer, with $\sigma(\cdot)$ as the non-linear activation function (e.g., ReLU), is:

$$\mathbf{x}^{(l)} = \sigma(\tilde{\mathbf{A}}\mathbf{x}^{(l-1)}\mathbf{w}^{(l)})$$

where $\tilde{\mathbf{A}} = \mathbf{D}^{-\frac{1}{2}}\mathbf{A}\mathbf{D}^{-\frac{1}{2}}$, where $\mathbf{D}$ is the node degree-matrix and $\mathbf{A}$ denoting the adjacency matrix. In RAP-GNN variants, all $\mathbf{w}^{(l)}$ are randomly sampled on-the-fly as diagonal matrices, as shown in Equation (1).

**GraphSAGE.** The forward pass formula for the $l$-th GraphSAGE layer is:

$$\mathbf{x}_v^{(l)} = \sigma\left(\mathbf{w}^{(l)} \cdot \text{AGGREGATE}\left(\{\mathbf{x}_v^{(l-1)}\} \cup \{\mathbf{x}_u^{(l-1)}, \forall u \in \mathcal{N}(v)\}\right)\right)$$

Where:

- $\mathbf{x}_v^{(l)}$ is the embedding of node $v$ at layer $l$-th, which is a component in $\mathbf{x}^{(l-1)}$.
- $\mathcal{N}(v)$ denotes the set of neighbors of node $v$.
- AGGREGATE is a neighborhood aggregation function (e.g., mean, LSTM, pooling).
- $\mathbf{w}^{(l)}$ is a trainable weight matrix at layer $l$, but with RAP-GNN variants, $\mathbf{w}^{(l)}$ are randomly sampled on-the-fly as diagonal matrices, as shown in Equation (1).
- $\sigma(\cdot)$ is a non-linear activation function.

## B  Discussion on Equivariance

Recall that each weight matrix $\mathbf{w}^{(l)}$ is randomly sampled at every forward pass. In practice, in all experiments in this paper, since each $\mathbf{w}^{(l)}$ is sampled for every forward pass, it is the same for all graphs in the same batch, but it is different for graphs in different batches. This implies that the output of RAP-GNN is permutation equivariant *for each graph*, but not across graphs. In other words, isomorphic nodes within each graph are assigned the same representation, while isomorphic nodes in different graphs may receive different representations.

This approach introduces a middle ground between permutation-equivariant GNNs, which assign the same representation to isomorphic nodes, regardless if they are in the same or in different graphs, and permutation-sensitive methods, such those employing RNF, which instead differentiate isomorphic nodes (with high probability). We believe that this hybrid approach makes it easier to yield correct predictions compared to RNF models, especially in the case of node classification task, as demonstrated by our experiments. Indeed, as isomorphic nodes are still assigned the same representation within a graph, the classifier does not need to map the different representations of the isomorphic nodes within a graph to the same prediction. We believe that properly investigating this aspect represents an interesting avenue for future research.

## C  Impact of Initial Embeddings

In this section, we examine how various configurations of the initial MLP-based embedding layer affect RAP-GNN's performance on the Cora, CiteSeer, and PubMed datasets. The primary role of this embedding layer is to extract meaningful features from the input data, enhancing overall model performance. To assess the impact of these configurations, we conduct an ablation study while keeping the GNN weights sampled on-the-fly, and majority voting scheme as described in Section 3. Specifically, we explore five distinct setups for sampling the MLP feature embedding weights.

For random sampling of $\phi$ in $h_\phi^{\text{pre}}$, the Identity Propagation Embedding (IPE), Fixed Propagation Embedding (FPE), and Dynamic Propagation Embedding (DPE) configurations apply the corresponding random sampling schemes to the GNN weights, as detailed in Section 4.1. For the learn-

Table 3: Ablation Study of MLP Embedding Configuration for Node Classification (↑) using GCN.

| Method ↓ / Dataset → | CORA | CITESEER | PUBMED |
|---|---|---|---|
| **NATURAL BASELINES** | | | |
| LW | 81.50 ± 4.8 | 71.10 ± 0.7 | 79.00 ± 0.6 |
| **RAP-GNN VARIANTS** | | | |
| DPW + IPE + RNF | 78.44 ± 2.95 | OOM | 73.46 ± 1.11 |
| DPW + FPE + RNF | 82.82 ± 0.95 | 71.48 ± 0.73 | 78.76 ± 0.55 |
| DPW + DPE + RNF | 26.30 ± 6.08 | 25.22 ± 0.85 | 37.28 ± 1.36 |
| DPW + LEARNEMB + RNF | **84.36 ± 0.29** | **72.16 ± 0.57** | **79.32 ± 0.26** |
| DPW + PREEMB + RNF | 83.33 ± 0.19 | 70.80 ± 0.32 | 78.70 ± 0.35 |

able embedding configurations, we evaluate two setups: (1) Learnable Embedding (LEARNEMB), where the MLP weights are learned alongside the classifier while maintaining dynamic random propagation in the GNN, and (2) Pretrained Embedding (PREEMB), where the MLP embedding is pretrained without the GNN and remains fixed during the main training phase (as outlined in Section 3). These setups are compared to the natural baseline model with fully learnable weights (LW), where all parameters are learnable during training.

The results in Table 3 offer valuable insights into the different configurations. Among the random propagation methods, the DPE setupwhere both MLP and GNN layers have random weightsstruggles to learn across all datasets, indicating that randomization in both layers hinders effective feature learning. Similarly, the IPE setup (excluding the MLP) causes significant performance drops on Cora and PubMed and leads to Out-of-Memory (OOM) issues on CiteSeer, which has more features, underscoring the impracticality of this approach for larger-scale datasets.

In contrast, the FPE configuration, where the MLP weights are fixed after initialization, shows improvements on Cora and CiteSeer but provides no significant gains on PubMed. This can be attributed to the same limitations found with Fixed Propagation in GNN layers, as discussed in Section 4.1, due to insufficient propagation. Nonetheless, fixed MLP weights still enable the model to learn meaningful features when paired with random GNN weights.

The LEARNEMB combined with DPW configuration demonstrates substantial gains across all datasets, even surpassing the baseline GCN. However, this setup requires backpropagation through the entire network, despite only updating the MLP embedding and classifier, increasing training complexity and reducing scalability. Nonetheless, the strong performance shows that a learnable embedding is essential, as random sampling at this stage fails to extract meaningful features, setting the upper bound for all variants using DPW.

To reduce training complexity, the PREEMB setup provides a balanced option. While it is slightly less accurate than the learnable embedding, it still outperforms the GCN baseline on Cora. However, its performance on CiteSeer and PubMed is lacking, despite its superior training time as shown in Table 2, indicating potential areas for improvement. Nonetheless, we believe the combination of PREEMB with DPW presents promising opportunities for further exploration.

# D  Further Evaluation

## D.1  Evaluation on Graph Classification Tasks

In this section, we extend our evaluation to a different taskgraph classificationusing a new GNN backbone, GIN [33], on TUDatasets [32]. We compare IPW, FPW, and DPW combined with learnable embedding (LEARNEMB) and DPW combined with pretrained embedding (PREEMB) against the natural baseline (LW), which trains the network end-to-end. We also included a network training end-to-end using RNF, LW + RNF [18], as we also incorporate RNF in the proposed method.

The results in Table 4 indicate that the variants of our proposed method, RAP-GNN, consistently achieve competitive accuracy compared to networks trained end-to-end with fully learnable weights. Notably, the combination of DPW with LEARNEMB either outperforms or matches the performance across all datasets. This demonstrates that RAP-GNN, despite not requiring training of all GNN weights in the hidden layers, can still achieve comparable results to end-to-end models. These find-

Table 4: Graph classification accuracy (%) ↑ on TUDatasets.

| Method ↓ / Dataset → | MUTAG | PTC | PROTEINS |
|---|---|---|---|
| **NATURAL BASELINES** | | | |
| LW + RNF [18] | $90.8 \pm 4.8$ | $64.4 \pm 6.7$ | $74.1 \pm 2.6$ |
| LW + LEARNEMB + RNF | $89.9 \pm 6.4$ | $62.5 \pm 6.9$ | $\mathbf{76.7 \pm 4.1}$ |
| **RAP-GNN VARIANTS** | | | |
| IPW + LEARNEMB + RNF | $91.0 \pm 4.7$ | $61.9 \pm 7.6$ | $74.6 \pm 4.0$ |
| FPW + LEARNEMB + RNF | $91.0 \pm 4.1$ | $61.8 \pm 10.6$ | $75.3 \pm 3.9$ |
| DPW + LEARNEMB + RNF | $\mathbf{92.5 \pm 2.6}$ | $\mathbf{65.2 \pm 5.3}$ | $76.5 \pm 4.9$ |
| DPW + PREEMB + RNF | $89.9 \pm 5.5$ | $64.0 \pm 4.6$ | $76.3 \pm 6.0$ |

Table 5: Node classification accuracy (%) ↑ with GraphSage backbone.

| Method | OGBN-ARXIV |
|---|---|
| **NATURAL BASELINES** | |
| LW [34] | $\mathbf{73.08 \pm 0.1}$ |
| **RAP-GNN VARIANTS** | |
| DPW + LEARNEMB + RNF | $70.36 \pm 0.2$ |
| DPW + PREEMB + RNF | $69.83 \pm 0.3$ |

ings reinforce the potential of RAP-GNN to enhance both accuracy and computational efficiency across various GNN architectures and tasks. However, similar to the results shown in Section 4.1 for node classification, there is a slight performance decrease when comparing DPW combined with LEARNEMB and PREEMB, highlighting it as a promising strategy for scenarios where scalability and efficiency are critical without sacrificing performance.

## D.2 Evaluation with different GNN backbones

We further evaluate the performance of RAP-GNN using another GNN backbone, GraphSage [5], on the large-scale dataset ogbn-arxiv. In all experiments, the feature embedding consists of a 1-hidden GNN layer, utilizing the same GraphSage backbone as the GNN hidden layers. We compare the combinations of DPW with LEARNEMB and PREEMB with RNF against networks trained end-to-end with the GraphSage backbone.

The results of this experiment are presented in Table 5. Consistent with the results observed using the GCN backbone in Appendix D.1, both combinations of DPW with LEARNEMB and PREEMB with RNF show only a small gap in accuracy, although they slightly lag behind the natural baseline trained end-to-end.

Although these results are still in the early stages and require further refinement, they provide strong empirical evidence for the effectiveness and efficiency of using random weights in GNN hidden layers, compared to traditional networks that require all components to be learned.

## D.3 Accuracy Analysis Using Majority Voting

In this section, we present the accuracy results with and without majority voting during the evaluation phase for the Cora and ogbn-arxiv datasets. The corresponding run times are shown in Table 2 in Section 4.

As seen in Table 6, for the Cora dataset, the accuracy with M=1 (without majority voting) is comparable to that with majority voting (M=5) and still surpasses the end-to-end trained networks. On the ogbn-arxiv dataset, the accuracy remains the same with (M=3) and without (M=1) majority voting, although it does not match the performance of end-to-end trained networks.

When we consider the run times from Table 2 in Section 4 with and without majority voting on both datasets, it becomes clear that omitting majority voting (setting M=1) significantly reduces inference time while maintaining comparable accuracy. This suggests that further investigation is needed to determine whether majority voting should be retained or dropped for better efficiency.

Table 6: Node classification accuracy (%)↑ for Cora and ogbn-arxiv with (M>1) and without (M=1) majority voting. The results show that without majority voting (M=1) achieves accuracy *comparable* to using majority voting.

| Method ↓ / Dataset → | CORA | | OGBN-ARXIV | |
|---|---|---|---|---|
| | M=5 | M=1 | M=3 | M=1 |
| **NATURAL BASELINES** | | | | |
| LW | – | $81.50 \pm 0.8$ | – | $73.41 \pm 0.2$ |
| LW + LEARNEMB + RNF | – | $81.25 \pm 0.6$ | – | $73.28 \pm 0.3$ |
| **RAP-GNN VARIANTS** | | | | |
| IPW + LEARNEMB | $\mathbf{82.28 \pm 0.6}$ | $81.72 \pm 0.2$ | $\mathbf{70.57 \pm 0.1}$ | $70.28 \pm 0.1$ |
| IPW + LEARNEMB + RNF | $81.86 \pm 0.5$ | $\mathbf{81.86 \pm 0.2}$ | $70.59 \pm 0.2$ | $\mathbf{70.63 \pm 0.2}$ |
| FPW + LEARNEMB | $81.92 \pm 0.9$ | $\mathbf{82.32 \pm 0.3}$ | $\mathbf{70.28 \pm 0.6}$ | $70.01 \pm 0.5$ |
| FPW + LEARNEMB + RNF | $82.34 \pm 0.8$ | $\mathbf{82.62 \pm 0.4}$ | $\mathbf{70.15 \pm 0.4}$ | $69.98 \pm 0.6$ |
| DPW + LEARNEMB | $\mathbf{82.82 \pm 0.9}$ | $82.52 \pm 0.6$ | $70.58 \pm 0.3$ | $\mathbf{70.84 \pm 0.3}$ |
| DPW + LEARNEMB + RNF | $\mathbf{84.36 \pm 0.3}$ | $83.16 \pm 0.6$ | $\mathbf{71.23 \pm 0.2}$ | $70.92 \pm 0.4$ |
| DPW + PREEMB + RNF | $\mathbf{83.33 \pm 0.2}$ | $83.07 \pm 0.5$ | $\mathbf{69.84 \pm 0.4}$ | $69.23 \pm 0.1$ |

Table 7: Runtime for pretraining (PRETRAIN) and training (TRAIN). The total time for both pretraining and training phases with RAP-GNN is *shorter* than that of end-to-end training methods.

| Method ↓ / Dataset → | CORA (ms) | | OGBN-ARXIV (ms) | |
|---|---|---|---|---|
| | PRETRAIN | TRAIN | PRETRAIN | TRAIN |
| **NATURAL BASELINES** | | | | |
| LW + LEARNEMB | – | 4.89 | – | 438.78 |
| LW + LEARNEMB + RNF | – | 5.10 | – | 441.63 |
| **RAP-GNN VARIANTS** | | | | |
| DPW + PREEMB + RNF | 1.09 | **2.04** | 41.60 | **228.59** |

## D.4 Training Time Analysis for RAP-GNN with PREEMB embeddings

In this section, we present the average runtime per epoch during the pretraining phase for the Cora and ogbn-arxiv datasets, as shown in Table 7. For smaller datasets like Cora, the time-saving benefits of using RAP-GNN combined with a pretrained embedding are less pronounced, as the total time for both the pretraining and training phases (as shown in Table 2, Section 4) is only slightly shorter than that of end-to-end trained networks. However, for larger datasets like ogbn-arxiv, the difference becomes significantly more substantial. The combined pretraining and training time is over half that of end-to-end training, leading to considerable time savings. This underscores the scalability and efficiency of RAP-GNN with a pretrained embedding for graph learning tasks.

## E Experiment Details

Our experiments were conducted using the PyTorch [35] and PyTorch Geometric [36] frameworks, utilizing WandB [37] for hyperparameter sweeps. In this section, we provide details on our specific implementation of the experiments.

Table 8 outlines the hyperparameter search space for all datasets. Note that all $\mathbf{w}^{(l)}$ matrices must be square, which requires that the input and output dimensions of all hidden layers in $g_{\mathbf{W}}$ are the same, and are all equal to HIDDEN DIM.. We use L to denote the number of hidden layers in $g_{\mathbf{W}}$. For Cora, CiteSeer, PubMed, and all datasets under TUDatasets, the number of MLP layers for the embedding is treated as a hyperparameter, while the hidden and output dimensions of the MLP are set equal to HIDDEN DIM.. All hyperparameter search details are provided Table 8.

**Node Classification Tasks.** For the Cora, CiteSeer, and PubMed datasets, all networks use a GCN backbone. The LW implementation follows the PyTorch Geometric example, incorporating Jumping

Table 8: Hyperparameters search for all models in different datasets, where L is the number of hidden layers in $g_{\mathbf{W}}$ and HIDDEN DIM. denotes the dimension of all hidden layers in $g_{\mathbf{W}}$.

| Dataset | #LAYER in $h_\phi^{\text{pre}}$ | L | LEARNING RATE | HIDDEN DIM. | #EPOCHS | BATCH SIZE | DROPOUT | #RNF |
|---|---|---|---|---|---|---|---|---|
| **Node Class.** | | | | | | | | |
| CORA | $\{1,2\}$ | $\{2,4,8,16\}$ | $\{0.01, 0.007, 0.005, 0.001\}$ | $\{16,32\}$ | $\{700\}$ | – | $\{0, 0.25, 0.4, 0.5\}$ | $\{0,2,4,6,8\}$ |
| CITESEER | $\{1,2,3\}$ | $\{2,4,6\}$ | $\{0.01, 0.007, 0.005, 0.001\}$ | $\{16,32,64,128\}$ | $\{1000\}$ | – | $\{0, 0.25, 0.4, 0.5\}$ | $\{0,2,4,6\}$ |
| PUBMED | $\{1,2,3\}$ | $\{4,6,8,12\}$ | $\{0.01, 0.007, 0.005, 0.001\}$ | $\{16,64,128,256\}$ | $\{1000\}$ | – | $\{0, 0.25, 0.4, 0.5\}$ | $\{0,4,8,12,16\}$ |
| OGBN-ARXIV | $\{1\}$ | $\{5,6,7,8\}$ | $\{0.05, 0.01, 0.005, 0.001\}$ | $\{128,256\}$ | $\{2000\}$ | – | $\{0, 0.25, 0.5, 0.6\}$ | $\{0,4,6,8\}$ |
| **Graph Class.** | | | | | | | | |
| MUTAG | $\{1,2,3\}$ | $\{1,2,3\}$ | $\{0.05, 0.03, 0.01, 0.007\}$ | $\{16,64,128\}$ | $\{1000\}$ | $\{64,128,512\}$ | $\{0.2\}$ | $\{0,2,4,6\}$ |
| PTC | $\{1,2,3\}$ | $\{2,3,4,6\}$ | $\{0.05, 0.03, 0.01, 0.007\}$ | $\{16,64,128\}$ | $\{1000\}$ | $\{64,128,512\}$ | $\{0.2\}$ | $\{0,2,4,8,12\}$ |
| PROTEINS | $\{1,2,3\}$ | $\{1,2,3,4\}$ | $\{0.05, 0.03, 0.01, 0.007\}$ | $\{16,32,64\}$ | $\{1000\}$ | $\{128,256,512\}$ | $\{0.2\}$ | $\{0,2,4,8,12\}$ |

Knowledge (JK) [38]. All variants of RAP-GNN are built on this foundation, using MLP-based embeddings. We set M=5 for the results reported with the Cora, CiteSeer, and PubMed datasets.

For the ogbn-arxiv dataset, the implementation is based on [34], featuring both GCN and GraphSage backbones. As mentioned, the embedding shares the same backbone as the GNN layers, with just one hidden layer (no further tuning). All variants of RAP-GNN are implemented on this structure. We utilize 3 voters for the results reported with the ogbn-arxiv dataset.

Moreover, in all node classification tasks, batching is not required for either training or evaluation; therefore, the batch size column in Table 8 for all datasets under node classification tasks is left blank.

Furthermore, for all considered experiments in all node classification tasks, we show the mean $\pm$ std. of 5 runs with different random seeds.

**Graph Classification Tasks.** For the TUDatasets, the LW implementation follows the PyTorch Geometric example, using GIN as the backbone. The three datasets used for evaluation are MUTAG, PROTEINS, and PTC_MR (PTC). All variants of RAP-GNN build on this foundation, again using MLP-based embeddings. We use 5 voters for the results presented in Table 4.

For evaluation, we followed the method used in Xu et al. [33]. For each dataset, we report the mean $\pm$ standard deviation of the validation accuracies across 10 folds in the cross-validation.

