# OpenReview forum: "Random Propagations in GNNs"
_NeurIPS.cc/2024/Workshop/UniReps — UniReps_

### Official Review · Reviewer_sUiX · 2024-10-03

**Rating:** 6
**Confidence:** 4

**Review:**

Summary:
The paper is an extended abstract on random propagation in graph neural networks (GNNs). It provides a background on GNN training efficiency methods and proposes Random Propagation GNN (RAP-GNN), a framework aimed at addressing the computational inefficiencies of GNNs. By utilizing random propagations, the model avoids end to end training like backpropagation, reducing training time by around 50% (as claimed in experiments) while retaining accuracy for node and graph classification tasks on small datasets like Cora, Citeseer, arxiv, etc. The architecture uses random weights for GNN layers while only training the final classifier.

Strengths:
- The use of random weights provides significant efficiency gain from its obvious framing w.r.t. backpropagation based training
- The performance of the RAP-GNN still is competitive despite only the final classifier being trained alongside the pretrained embeddings at the input.
- The ablation studies provide investigation on ranges of random strategies from identity to dynamic random weight sampling; although there is no single type which works best in all datasets

Limitations:
- minor: some references are repeated such as Kipf and Welling's for GCN
- one major limitation of this work is its missing evaluation on datasets which are more large and are used in standard works on GNNs that study scalability and efficiency (such as ogb products or similar scale of millions nodes). demonstrating how the random weights provide accuracy retention and efficiency gains on such scale would be a better demonstration of the utility of RAP-GNN. At the same time, the smaller datasets such as Cora, etc. could be inappropriate due to their rich features on the nodes which does not necessarily warrant powerful GNNs (incl. backprop trained)
- although RAP-GNN reduces training time considerably, the inference process is more complex due to the need for majority voting (M > 1). This increases inference time, particularly when compared to models that do not require such redundancy. as such, this could become a bottleneck in applications where real-time inference is critical.

---

### Official Review · Reviewer_tqcF · 2024-10-04
**Nice and interesting investigation of random initilaztion of GNN weight**

**Rating:** 8
**Confidence:** 4

**Review:**

# Summary
It is interesting to read this paper as intuitively the random initialization does not make sense for a traditional graph training pipeline. However, the authors proposed a framework that utilizes dynamically sampled random weights in GNN layers combined with a pretrained feature embedding, reducing the need for backpropagation through every layer. This approach requires training only the final classifier, resulting in a **50% reduction in training time** while **maintaining competitive accuracy** in node and graph classification tasks.

# Strength
1. It significantly reduces computational costs by avoiding backpropagation in the GNN layers, focusing only on training the final classifier. This enables faster training, as evidenced by a 50% reduction in training time with competitive accuracy in both node and graph classification tasks compared to traditional GNNs.
2. It explains the motivation behind the random initialization from the random models and graph lottery-ticket hypothesis. I believe this paper can further discuss the relevance of papers for **adding noise to the features [1]** and **removing graph neural networks during training [2,3]**. If the random initialization works well for different graph tasks, it is useful to regard the technique as a data augmentation process or the graph convolution itself is not important in training.
3. The presentation is clear.

# Weakness
1. The random propagation term sounds confusing, as one would expect the features to be randomly propagated via different random edges during the graph convolution process. However, it is mainly about the random initialization of the GNN weight during graph convolution.
2. Comprehensive experiments on more datasets (more large-scale graph which values more on training time) and tasks (link prediction) with different graph structures (more than gcn, gin, and graphsage) are needed to validate this idea.


[1] Yu J, Yin H, Xia X, et al. Are graph augmentations necessary? simple graph contrastive learning for recommendation[C]//Proceedings of the 45th international ACM SIGIR conference on research and development in information retrieval. 2022: 1294-1303.

[2] Zhang W, Yang L, Song Z, et al. Do We Really Need Graph Convolution During Training? Light Post-Training Graph-ODE for Efficient Recommendation[J]. arXiv preprint arXiv:2407.18910, 2024.

[3] Ju M, Shiao W, Guo Z, et al. How Does Message Passing Improve Collaborative Filtering?[J]. arXiv preprint arXiv:2404.08660, 2024.

---

### Official Review · Reviewer_ESJ7 · 2024-10-05
**Review of Submission #53**

**Rating:** 6
**Confidence:** 4

**Review:**

Paper Summary:  This paper tackles the problem of resource efficient training of Graph Neural Networks on Large Graphs via the use of Random Propagations for training. Specifically, their Random Propagation GNN (RAP-GNN) is evaluated on whether random propagations in GNNs are as effective as end-to-end optimisation for downstream tasks and whether random weights, requiring only training of a final classifier head are sufficiently performant.

Strengths:
- Various Randomness Strategies are tested and appropriate baselines have been reported.
- Different datasets of varying sizes are used for testing
- Introduction of Randomness through does not degrade performance significantly in the experiments
- Computational cost is consistently reduced by using the random propagation approach

Weaknesses:
- It is not clear how much data is required for pre-training the joint classifier and GNN head and what the computation overhead here is. How much does performance depend on pre-training data quality?
- Experiments are restricted to classification settings. The discussion on permutation equivariance in the appendix raises questions on generalisability to tasks outside of node and graph-classification, eg. link prediction/regression etc

Questions:
- It is not very clear how many independent voter models are used in Table 1 and 3 (appendix)

---

### Official Review · Reviewer_2Wu5 · 2024-10-06
**Interesting method to save computation when training GNNs**

**Rating:** 6
**Confidence:** 4

**Review:**

In this paper the authors present a method to train a GNN without having to perform backpropagation through the whole model. By using a pretrained embedding layer, randomly sampled weights and a decoder at the end, the method is able to beat fully learnable weights in some benchmarks.

The paper presents good results, but is missing results from using a single voter count (i.e., $M = 1$) in Table 1 in order to substantiate the argument that the increase in inference time compute is necessary to beat the performance over the baselines.

---

### Decision · Program_Chairs · 2024-10-10

**Decision:**

Accept

**Comment:**

In light of the positive reviewers' feedback and relevancy of the submission, we are pleased to accept this paper for presentation at UniReps 2024. We kindly ask the authors to incorporate the reviewers' suggestions and feedback in the final camera-ready version of the manuscript.